# Predicting Institution Hierarchies with Set-based Models

**Derek Tam**                                                 DPTAM@CS.UMASS.EDU
*University of Massachusetts Amherst*

**Nicholas Monath**                                       NMONATH@CS.UMASS.EDU
*University of Massachusetts Amherst*

**Ari Kobren**                                         ARI.KOBREN@ORACLE.COM
*Oracle Labs*
*University of Massachusetts Amherst*

**Andrew McCallum**                                     MCCALLUM@CS.UMASS.EDU
*University of Massachusetts Amherst*

## Abstract

The hierarchical structure of research organizations plays a pivotal role in science of science research as well as in tools that track the research achievements and output. However, this structure is not consistently documented for all institutions in the world, motivating the need for automated construction methods. In this paper, we present a new task and model for predicting sub-institution/super-institution relationships based on their string names. The crux of our model is that it leverages learned, permutation invariant representations of various token subsets of institution name strings. Our model outperforms or matches non-set-based models and baselines. We also create a dataset for training and evaluating models for this task based on the publicly available relationships in the Global Research Identifier Database.

## 1. Introduction

Academic, government and various industrial organizations are often hierarchically structured [Fan et al., 2012]. For example, the *Canadian Institute for Theoretical Astrophysics* is a sub-institution of *University of Toronto* and *Harvard University* is the super-institution of both *Harvard Medical School* and *Harvard Divinity School*.

Science of science research focuses on interactions among various scientific agents (researchers, academic institutions, etc.) with the goal of developing tools and policies for accelerating science [Fortunato et al., 2018]. Institution hierarchies are used in the analysis of the funding of institutions and, when accompanied with temporal information, organizational hierarchies can also provide insights into the evolution / growth of institutions as well as fine-grained information about the mobility of researchers, capturing department and career focus changes [Azoulay et al., 2011, Stephan, 2012]. Thus, correctly modeling organizational hierarchies is critical for tool development in science of science and other institutional research.

The Global Research Identifier Database (GRID) is a resource meant to support such efforts. GRID is carefully curated and contains detailed information about research institutions, includ-

ing their sub-institution/super-institution relationships. While the database is large and growing (containing almost 100k institutions at the time of writing), there are institutions and relationships that are missing. For example, *Montreal Institute for Learning Algorithms* is not present in GRID, and the public hospital, *John Hunter Children's Hospital*, in New South Wales is not connected to the Government of New South Wales nor Australia's Government / healthcare system. Similarly, fine grained information, such as particular research laboratories or smaller departments within universities' colleges, are often absent.

Relation extraction methods are typically used to create knowledge-bases (KBs) containing facts such as which institutions are parents/children of one another. OpenIE [Etzioni et al., 2011, Fader et al., 2011, inter-alia] and Universal Schema [Riedel et al., 2013, Verga et al., 2016, Das et al., 2017, inter-alia], are able to extract complex relationship between entities from text. These methods are either based on link-prediction techniques or use rich contextual information to make accurate predictions. Unfortunately, such context is often unavailable. For example, funding agencies listed in research papers, assignees mentioned in patents, and author/inventor affiliations included in articles and BibTeX entries regularly appear as strings with little to no context relevant to making predictions about organization hierarchies. Access to a KB of institutions names and other features may be available, but these KBs often miss hierarchical relationships between those institutions.

In this paper, we explore methods for predicting hierarchical relationships between institutions based solely on the spelling of the institution names and knowledge-base features. While some sub-institution/super-institution relationships can be predicted using simple token overlap between institution strings (e.g., *Harvard Medical School* `sub-institution-of` *Harvard University*), others require methods which can perform richer semantic understanding beyond simple token matching (e.g., *Centre d'Investigation Clinique Pierre Drouin,Vandœuvre-lès-Nancy, France* `sub-institution-of` *Centre Hospitalier Universitaire de Nancy,Nancy, France*). Predicting sub-institution/super-institution relationships without additional context is important when trying to complete KBs such as GRID without intensive data gathering for contextual evidence.

We present a model that predicts `sub-institution-of` relationships using institution names, locations, and types. Inspired by the intuition that set-intersections and set-differences among the tokens of two institution names provide significant signal as to their relationship, our model learns to combine permutation invariant representations of token subets derived from institution names. Operating over embedded representations improves generalizability in comparison to techniques that operate on discrete tokens.

In this work, we make the following 3 contributions: (1) we present a new **set-based model** that outperforms non-set-based models on sub-institution/super-institution relationship prediction, (2) we **examine the difficulty** of the task and show when set-based models work well, and (3) we create and release a **new dataset** derived from GRID to support sub-/super-institution relationship prediction.

## 2. Background

### 2.1 Dataset & Preprocessing

The Global Research Identifier Database (GRID), contains information about 91,640 institutions from 217 countries[1]. The institutions include universities, government agencies, companies, health

---

1. We use the 2018-11-14 Release.

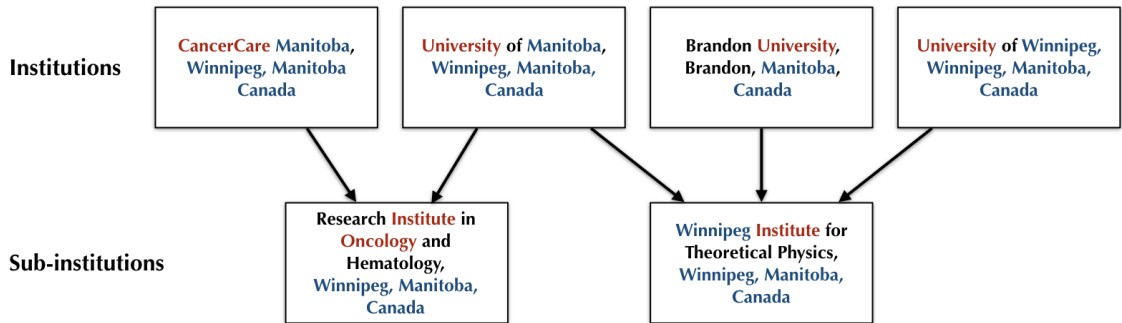

Figure 1: **GRID Institution Hierarchy**. A connected component in the GRID institution graph. Tokens highlighted in blue are members of the intersection of the sub-institution and super-institution strings. Tokens highlighted in red are not a members of this intersection, however they are semantically conveyed in both the the sub-institution and super-institution strings.

care organizations and other places of research. GRID stores the name, location, and type (university, hospital, government agency, etc), as well as the parent of each institution. The database includes 11,393 instances of the sub-institution relationship.

Our goal is to develop models that predict the hierarchical structure of institutions. We use GRID to derive a new dataset that contains pairs of institutions that participate in the sub-institution relationship (i.e., positive examples) and pairs of institutions that do not (i.e., negative examples). Constructing our dataset begins by building a graph where each node represents a GRID institution, and each edge represents a parent-child relationship among an institution pair. We partition the child-parent edges into a train/dev/test split by randomly assigning each connected component of institutions and all of its edges to either the train, development or test set (using a $0.6/0.2/0.2$ split). We construct negative examples in three ways: (1) non-sub-institutions in the same connected component, (2) non-sub-institutions in the same city in a different connected component, and (3) randomly sampled pairs. Figure 1 visualizes a connected component from GRID. Due to its overwhelming size (2981 sub-institutions) with respect to other institutions in GRID, we exclude the connected component representing the United States Government from the dataset.

### 2.2 Institution Hierarchy Prediction

Let $A$ and $B$ be two institutions in GRID. Then, $A$ `sub-institution-of` $B$, if $A$ is a descendant of $B$ according to GRID. In institution hierarchy prediction, we are given two GRID institutions, $A$ and $B$, which includes their names, locations, and types, and asked to predict if the $A$ `sub-institution-of` $B$.

## 3. Set-based Models for Predicting Institution Hierarchies

In this section, we present a model for predicting whether one institution is a sub-institution of another. Our model makes use of Set Transformers [Lee et al., 2019] to model the token overlap between the institution names.

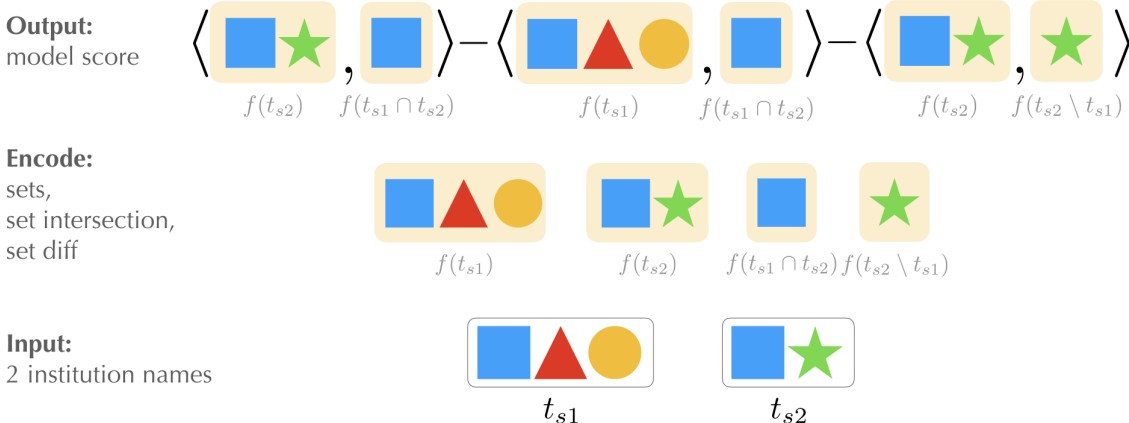

**Output:**
model score

$f(t_{s2})$ $f(t_{s1} \cap t_{s2})$ $f(t_{s1})$ $f(t_{s1} \cap t_{s2})$ $f(t_{s2})$ $f(t_{s2} \setminus t_{s1})$

**Encode:**
sets,
set intersection,
set diff

$f(t_{s1})$ $f(t_{s2})$ $f(t_{s1} \cap t_{s2})$ $f(t_{s2} \setminus t_{s1})$

**Input:**
2 institution names

$t_{s1}$ $t_{s2}$

Figure 2: **Set-based predictor**. Given two institution names represented as sets of tokens, the predictor embeds the sets, their intersection, and set difference. It uses these embeddings to compute a score representing the likelihood that the first institution is a sub-institution of the second.

### 3.1 Sets of Institution Tokens

When predicting whether one institution is a sub-institution of another, the collection of tokens that appear in *both* institution names constitutes a strong signal. For example, consider the pair of institutions *University of Pennsylvania* and *The Wharton School of the University of Pennsylvania*. In this case, one institution name is a substring of the other, suggesting that in such cases the substring-institution name is a super-institution of the other.

However, this heuristic is far too strict. For example, consider the institution pair *The Ohio State University* `sub-institution-of` *University System of Ohio*, where the substring relationship does not hold. In this example, we notice that the two strings share the tokens {*Ohio, University*}, though the tokens appear in different orders in the strings. This suggests treating institution names as *unordered sets* of tokens rather than ordered strings. Additionally, the most salient tokens appear in both institution names and less relevant tokens do not. This suggests that it is more likely for some tokens to appear in both names if two institutions participate in the `sub-institution-of` relationship than others.

We present a model inspired by these insights, which is designed to predict whether one institution is a sub-institution of another. The model takes an ordered pair of institution names as input, $(s_1, s_2)$, and outputs a score: $h(s_1, s_2)$, with larger scores corresponding to greater model confidence in $s_1$ `sub-institution-of` $s_2$. We use $t_s$ be the tokens of a string $s$. The model is a function of four sets, each of which can be built from the two institution names: (1) the first institution represented as a set of tokens ($t_{s_1}$), (2) the second institution represented as a set of tokens ($t_{s_2}$), (3) the intersection between the first and second institution ($t_{s_1} \cap t_{s_2}$), and (4) the set difference between the second institution and the first, ($s_2 \setminus s_1$).

Let $f : 2^V \to \mathbb{R}^d$ be a function that returns the embedded representation of a token set, where $V$ is the vocabulary defined by our model. Define the model, $h$, evaluated on two string, $s_1$ and $s_2$, as

follows:

$$h(s_1, s_2) = \langle f(t_{s_2}), f(t_{s_2} \cap t_{s_1}) \rangle - \langle f(t_{s_1}), f(t_{s_2} \cap t_{s_1}) \rangle - \langle f(t_{s_2}), f(t_{s_2} \setminus t_{s_1}) \rangle \qquad (1)$$

where $\langle \cdot, \cdot \rangle$ is the inner product of the two vectors.

The first term in Equation 1 computes the dot product between $s_2$, the candidate super-institution, and the intersection of the two institutions strings. As previously discussed, if all tokens of $s_2$ appear in $t_{s1} \cap t_{s2}$, then it is likely that $s_1$ `sub-institution-of` $s_2$. The second term encodes a dual effect: if the embedded representation of $s_2$, i.e., $f(t_2)$ is similar to the embedding of the intersection of the tokens of $s_1$ and $s_2$, i.e., $f(t_{s_1} \cap t_{s_2})$, then $s_2$ is likely a super-institution of $s_1$ (captured in the subtraction of the second term from the first). Together, the first two terms encode the intuition that children institutions tend to exhibit a higher degree of specificity than their parents. Finally, the third term in the model compares the tokens in $s_2$ (including tokens in the intersection) to the tokens which are *unique* to $s_2$. If there is large overlap between these sets, it indicates that $s_2$ is significantly different from $s_1$, i.e., $s_1$ is unlikely to be a sub-institution of $s_2$. The third term in Equation 1 compares the tokens in $s_2$ (and potentially in the intersection) to the tokens which are *unique* to $s_2$. A large overlap between these sets indicates that $s_2$ is sufficiently different from $s_1$, i.e., $s_1$ is unlikely to be a sub-institution of $s_2$. Figure 2 visualizes our model. Any set encoder function may be used for $f$. Our choice of this function is discussed further in Section 3.2.

### 3.1.1 TRAINING OBJECTIVE

We train our model to optimize a ranking objective. Given a triple of institutions $(D, A_{pos}, A_{neg})$ such that $D$ `sub-institution-of` $A_{pos}$ is True and $D$ `sub-institution-of` $A_{neg}$ is False, we use the Bayesian Personalized Ranking objective as our loss function $\sigma(h(D, A_{pos}) - h(D, A_{neg}))$ [Rendle et al., 2009].

### 3.2 Set Encoders

Our model relies on the function, $f$, to provide a meaningful representations of sets of institution name tokens. While many permutation invariant, set-based models haven been proposed in recent work, [Ravanbakhsh et al., 2016, Zaheer et al., 2017, Ilse et al., 2018, Hartford et al., 2018, Lee et al., 2019, Mena et al., 2018, Cotter et al., 2019, Bloem-Reddy and Teh, 2019], in this paper, we use a *Set Transformer* [Lee et al., 2019], which represents the state-of-the-art permutation invariant model based on transformers [Vaswani et al., 2017]. Given a set of tokens, the Set Transformer encodes the tokens without position embeddings. We take the set representation to be the average of the final representation after the last layer in the transformer for each token in the set. The size of the sets used are relatively small, and so we do not use the inducing point-based Set Transformers and instead use all pairwise self-attention [Lee et al., 2019].

### 3.3 Institution Features

In addition to the name, GRID contains the city, state, country, and type (i.e., Education, Government, Nonprofit) for each institution. To train our model, we apply $h$ (Eq. 1) to each feature and get a score for each feature. Then, the final score is learned weighted sum of the scores for each feature. Note that each feature has its own parameters. Institution name and city have a token vocabulary while the

| Model | Ablation | MAP | Acc. at 1 | Acc. at 10 | Acc. at 50 | Acc. at 100 |
|---|---|---|---|---|---|---|
| **TokSim** | Orig | 0.060 | 0.037 | 0.161 | 0.332 | 0.384 |
| **O-LSTM** | Orig | $0.166 \pm 0.04$ | $0.105 \pm 0.05$ | $0.331 \pm 0.04$ | $\mathbf{0.598 \pm 0.07}$ | $\mathbf{0.728 \pm 0.08}$ |
| | Unig | $0.170 \pm 0.02$ | $0.116 \pm 0.03$ | $0.329 \pm 0.05$ | $\mathbf{0.624 \pm 0.07}$ | $\mathbf{0.745 \pm 0.07}$ |
| **O-Trans** | Orig | $0.128 \pm 0.05$ | $0.086 \pm 0.04$ | $0.118 \pm 0.08$ | $0.484 \pm 0.05$ | $0.604 \pm 0.06$ |
| | Unig | $0.103 \pm 0.02$ | $0.050 \pm 0.02$ | $0.225 \pm 0.04$ | $0.535 \pm 0.08$ | $0.643 \pm 0.08$ |
| **S-AvgEmb** | Orig | $0.182 \pm 0.03$ | $0.133 \pm 0.06$ | $\mathbf{0.327 \pm 0.03}$ | $0.610 \pm 0.02$ | $0.701 \pm 0.02$ |
| | LC | $0.177 \pm 0.02$ | $0.11 \pm 0.02$ | $0.308 \pm 0.03$ | $0.598 \pm 0.03$ | $0.684 \pm 0.04$ |
| | MLP | $0.146 \pm 0.02$ | $0.077 \pm 0.02$ | $0.280 \pm 0.04$ | $0.605 \pm 0.01$ | $0.708 \pm 0.02$ |
| | Unig | $0.090 \pm 0.01$ | $0.022 \pm 0.01$ | $\mathbf{0.193 \pm 0.21}$ | $0.559 \pm 0.03$ | $0.684 \pm 0.02$ |
| **S-Trans (Ours)** | Orig | $\mathbf{0.262 \pm 0.07}$ | $\mathbf{0.248 \pm 0.11}$ | $\mathbf{0.407 \pm 0.07}$ | $\mathbf{0.630 \pm 0.09}$ | $\mathbf{0.714 \pm 0.10}$ |
| | LC | $0.128 \pm 0.06$ | $0.057 \pm 0.04$ | $\mathbf{0.281 \pm 0.23}$ | $\mathbf{0.573 \pm 0.10}$ | $\mathbf{0.696 \pm 0.07}$ |
| | LCI | $\mathbf{0.303 \pm 0.03}$ | $\mathbf{0.318 \pm 0.07}$ | $\mathbf{0.421 \pm 0.05}$ | $\mathbf{0.665 \pm 0.05}$ | $\mathbf{0.771 \pm 0.03}$ |
| | MLP | $0.170 \pm 0.05$ | $0.128 \pm 0.09$ | $0.321 \pm 0.05$ | $\mathbf{0.549 \pm 0.09}$ | $\mathbf{0.655 \pm 0.09}$ |
| | Unig | $0.135 \pm 0.01$ | $0.061 \pm 0.01$ | $0.254 \pm 0.02$ | $\mathbf{0.601 \pm 0.06}$ | $\mathbf{0.732 \pm 0.05}$ |

Table 1: Score using Institution and Location (city, state, country) and Type features

state, country and type have a character vocabulary (for the sake of generalization to unseen location strings).

## 3.4 Learning Model Coefficients

Note that the model $h$ (Eq. 1) can be rewritten as

$$h(s_1, s_2) = \langle f(t_{s_2}), f(t_{s_2} \cap t_{s_1}) \rangle - \langle f(t_{s_1}), f(t_{s_2} \cap t_{s_1}) \rangle - \langle f(t_{s_2}), f(t_{s_2} \setminus t_{s_1}) \rangle$$
$$= \left\langle \begin{bmatrix} 1 & -1 & -1 \end{bmatrix}, \begin{bmatrix} \langle f(t_{s_2}), f(t_{s_2} \cap t_{s_1}) \rangle & \langle f(t_{s_1}), f(t_{s_2} \cap t_{s_1}) \rangle & \langle f(t_{s_2}), f(t_{s_2} \setminus t_{s_1}) \rangle \end{bmatrix} \right\rangle.$$

To determine how important the specific combination of terms for our model $h$ is, we replace the vector $\begin{bmatrix} 1 & -1 & -1 \end{bmatrix}$ with a learnable weight vector, a learnable weight vector initialized randomly, and a learnable weight vector initialized with our proposed model (which we call the default initialization). This only represents a linear combination of the values

$$\begin{bmatrix} \langle f(t_{s_2}), f(t_{s_2} \cap t_{s_1}) \rangle & \langle f(t_{s_1}), f(t_{s_2} \cap t_{s_1}) \rangle & \langle f(t_{s_2}), f(t_{s_2} \setminus t_{s_1}) \rangle \end{bmatrix}.$$

## 4. Experiments

We evaluate the ability of our model to predict sub-institution relationships between institutions in a retrieval-based task on our GRID dataset. Given the name of an institution (query), we rank the candidate super-institution name. We evaluate model performance in terms of mean average precision (MAP) and accuracy at K (Acc.) [2] for K = 1, 5, 50, and 100 . The descendant institutions are randomly assigned to be queries in a train/dev/test split. We compare the following models:

- **SetTransformer (S-Trans)** - The proposed approach of this paper, our set-based model with a set transformer [Lee et al., 2019] for the function $f$.

---

2. We use the following measurement: number of sub-institutions in top K / min(K, number of sub-institutions)

| Model | Ablation | MAP | Acc. at 1 | Acc. at 10 | Acc. at 50 | Acc. at 100 |
|---|---|---|---|---|---|---|
| TokSim | Orig | 0.070 | 0.058 | 0.0192 | 0.389 | 0.481 |
| O-LSTM | Orig | $0.099 \pm 0.03$ | $0.044 \pm 0.02$ | $0.212 \pm 0.07$ | $0.506 \pm 0.08$ | $0.636 \pm 0.07$ |
| | Unig | $\mathbf{0.155 \pm 0.02}$ | $\mathbf{0.103 \pm 0.03}$ | $0.292 \pm 0.02$ | $0.546 \pm 0.04$ | $0.659 \pm 0.04$ |
| O-Trans | Orig | $0.079 \pm 0.01$ | $0.035 \pm 0.01$ | $0.174 \pm 0.01$ | $.382 \pm 0.03$ | $0.506 \pm 0.03$ |
| | Unig | $0.071 \pm 0.01$ | $0.034 \pm 0.01$ | $0.155 \pm 0.03$ | $0.353 \pm 0.03$ | $0.446 \pm 0.08$ |
| S-AvgEmb | Orig | $0.082 \pm 0.01$ | $0.042 \pm 0.01$ | $0.153 \pm 0.02$ | $0.378 \pm 0.02$ | $0.511 \pm 0.03$ |
| | LC | $0.097 \pm 0.01$ | $0.055 \pm 0.01$ | $0.167 \pm 0.02$ | $0.404 \pm .04$ | $0.528 \pm 0.03$ |
| | MLP | $0.081 \pm 0.01$ | $0.043 \pm 0.01$ | $0.147 \pm 0.02$ | $0.377 \pm 0.03$ | $0.512 \pm 0.05$ |
| | Unig | $0.045 \pm 0.01$ | $0.013 \pm 0.00$ | $0.019 \pm 0.01$ | $0.259 \pm 0.05$ | $0.381 \pm 0.07$ |
| S-Trans(Ours) | Orig | $\mathbf{0.192 \pm 0.05}$ | $\mathbf{0.126 \pm 0.07}$ | $\mathbf{0.382 \pm 0.04}$ | $\mathbf{0.657 \pm 0.04}$ | $\mathbf{0.750 \pm 0.03}$ |
| | LC | $\mathbf{0.175 \pm 0.05}$ | $\mathbf{0.132 \pm 0.08}$ | $\mathbf{0.312 \pm 0.05}$ | $0.546 \pm 0.04$ | $0.658 \pm 0.05$ |
| | LCI | $\mathbf{0.184 \pm 0.09}$ | $\mathbf{0.132 \pm 0.12}$ | $\mathbf{0.336 \pm 0.12}$ | $\mathbf{0.626 \pm 0.09}$ | $.734 \pm 0.07$ |
| | MLP | $\mathbf{0.129 \pm 0.02}$ | $\mathbf{0.070 \pm 0.01}$ | $0.249 \pm 0.05$ | $\mathbf{0.486 \pm 0.16}$ | $\mathbf{0.591 \pm 0.17}$ |
| | Unig | $0.119 \pm 0.03$ | $0.069 \pm 0.01$ | $0.208 \pm 0.05$ | $0.506 \pm 0.07$ | $0.649 \pm 0.07$ |

Table 2: Score using Institution and Location (city, state, country) features Only

| Model | Ablation | MAP | Acc. at 1 | Acc. at 10 | Acc. at 50 | Acc. at 100 |
|---|---|---|---|---|---|---|
| TokSim | Orig | 0.062 | 0.149 | 0.218 | 0.325 | 0.401 |
| O-LSTM | Orig | $0.087 \pm 0.02$ | $0.040 \pm 0.02$ | $\mathbf{0.198 \pm 0.07}$ | $\mathbf{0.488 \pm 0.07}$ | $\mathbf{0.601 \pm 0.08}$ |
| O-Trans | Orig | $0.067 \pm 0.02$ | $0.036 \pm 0.01$ | $0.144 \pm 0.06$ | $\mathbf{0.355 \pm 0.08}$ | $0.471 \pm 0.08$ |
| S-AvgEmb | Orig | $0.032 \pm 0.00$ | $0.011 \pm 0.00$ | $0.056 \pm 0.01$ | $0.235 \pm 0.02$ | $0.397 \pm 0.02$ |
| | LC | $0.035 \pm 0.00$ | $0.015 \pm 0.00$ | $0.061 \pm 0.01$ | $0.225 \pm 0.02$ | $0.406 \pm 0.02$ |
| | MLP | $0.034 \pm 0.00$ | $0.013 \pm 0.00$ | $0.013 \pm 0.00$ | $0.258 \pm 0.01$ | $0.410 \pm 0.02$ |
| S-Trans (Ours) | Orig | $\mathbf{0.135 \pm 0.02}$ | $\mathbf{0.077 \pm 0.01}$ | $\mathbf{0.237 \pm 0.06}$ | $0.478 \pm 0.07$ | $\mathbf{0.646 \pm .07}$ |
| | LC | $\mathbf{0.123 \pm 0.03}$ | $\mathbf{0.071 \pm 0.02}$ | $\mathbf{0.227 \pm 0.05}$ | $\mathbf{0.379 \pm 0.05}$ | $0.503 \pm 0.07$ |
| | LCI | $0.115 \pm 0.02$ | $\mathbf{0.064 \pm 0.02}$ | $0.214 \pm 0.04$ | $\mathbf{0.443 \pm 0.16}$ | $\mathbf{0.587 \pm 0.04}$ |
| | MLP | $\mathbf{0.156 \pm 0.01}$ | $\mathbf{0.099 \pm 0.02}$ | $\mathbf{0.276 \pm 0.02}$ | $\mathbf{0.427 \pm 0.02}$ | $0.520 \pm 0.04$ |

Table 3: Score using Institution Features Only

- **SetAverageEmbedding (S-AvgEmb)** - Our set-based model with a simple average of embeddings for the function $f$.
- **OrderedLSTM (O-LSTM)** - An order-based model that encodes each string using an LSTM and represents that string as the last LSTM hidden state. We use a dot product to score two encoded strings.
- **OrderedTransformer (O-Trans)** - An order-based model that represents each string as the average of the tokens encoded by a transformer. We use a dot product to score two encoded strings.
- **TokenSimilarity (TokSim)** - Rank candidate pairs by $\frac{|t_{\text{par}} \cap t_{\text{ch}}|}{|t_{\text{par}}|}$

We also experiment with several methods of combining the three terms of our set-based scoring function (Equation 1):

- **Original (Orig)** - the scoring function $h$ described above.

- **Linear Combination (LC)** - a learned linear combination of the values.

| Model | MAP | Acc. at 1 | Acc. at 10 | Acc. at 50 | Acc. at 100 |
|---|---|---|---|---|---|
| **TokSim** | $0.096 \pm 0.05$ | $0.074 \pm 0.04$ | $0.243 \pm 0.10$ | $0.389 \pm 0.05$ | $0.451 \pm 0.14$ |
| **O-LSTM** | $\mathbf{0.296 \pm 0.08}$ | $\mathbf{0.234 \pm 0.12}$ | $\mathbf{0.504 \pm 0.06}$ | $\mathbf{0.775 \pm 0.04}$ | $\mathbf{0.858 \pm 0.03}$ |
| **O-Trans** | $\mathbf{0.200 \pm 0.04}$ | $0.105 \pm 0.03$ | $\mathbf{0.416 \pm 0.05}$ | $\mathbf{0.734 \pm 0.03}$ | $\mathbf{0.826 \pm 0.02}$ |
| **S-AvgEmb** | $\mathbf{0.198 \pm 0.06}$ | $\mathbf{0.153 \pm 0.10}$ | $0.339 \pm 0.05$ | $0.655 \pm 0.02$ | $0.748 \pm 0.02$ |
| **S-Trans** | $\mathbf{0.293 \pm 0.01}$ | $\mathbf{0.277 \pm 0.06}$ | $\mathbf{0.453 \pm 0.03}$ | $\mathbf{0.686 \pm 0.07}$ | $\mathbf{0.762 \pm 0.10}$ |

Table 4: Cross Validation Scores

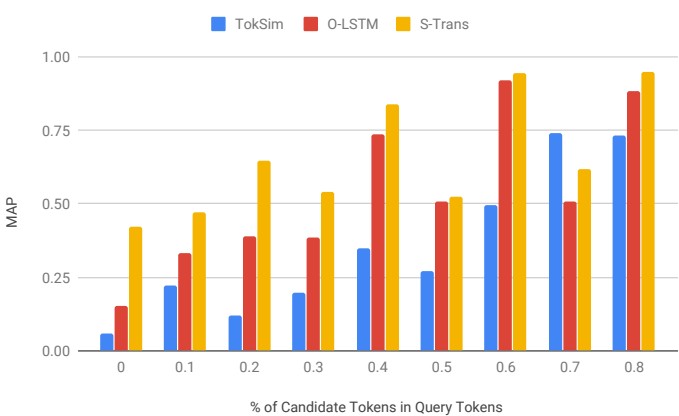

Figure 3: Performance by Token Overlap of Query-Candidate Institution Strings

- **Linear Combination Initialized (LCI)**, a learned linear combination of the values initialized as the scoring function $h$.

- **Multilayer Perceptron (MLP)** - a one layer multi-layer perceptron.

We also experiment with using a token vocabulary (**Unig**) for every feature (including the state, country, and type), rather than just the institution name and city as in our model.

For each model, we report the mean performance metric and its standard deviation across 5 different random seeds. We **bold** the model with the highest mean score, and any model whose mean is within 1 standard deviation. Table 1 shows the result for this experiment. We also compare the models when given access to subsets of the additional GRID features (i.e., location, type). Table 2 contains the results when given access to institution name and location. Table 3 contains the results when given access to institution name only.

We provide the code and data to reproduce our experimental results[3]. Models are implemented in PyTorch. The models are trained using Adam [Kingma and Lei Ba, 2015]. We perform hyperparameter tuning (embedding dimension, feed-forward layer dimension, number of heads, number of layers) on a development set [4] and report performance on the test set for the model which performed highest in terms of MAP on the development set.

---

3. https://github.com/iesl/institution_hierarchies
4. Hyperparameter search and configurations released in the code

| Descendant Entity | Prediction by our model | Prediction by OrderedLSTM |
|---|---|---|
| Northampton General Hospital, Northampton, England,United Kingdom | Northampton General Hospital NHS Trust, Northampton, United Kingdom | Northamptonshire Healthcare NHS Foundation Trust, Kettering, United Kingdom |
| NIHR Leeds Musculoskeletal Biomedical Research Unit, Leeds, United Kingdom | Leeds Teaching Hospitals NHS Trust, Leeds, United Kingdom | National Institute for Health Research Leeds, United Kingdom |

Table 5: **Comparison between our model and OrderedLSTM**. This table presents examples for which our model makes a correct prediction and OrderedLSTM makes an incorrect prediction. We observe that our model correctly utilizes location information.

| Descendant Entity | Prediction by our model | Prediction by TokenSim |
|---|---|---|
| The National Institute for Strategic Studies, Kiev,Ukraine | National Academy of Sciences of Ukraine,Kiev, Ukraine | Ukrainian Institute of Public Health Policy ,Kiev,Ukraine |
| NIHR Oxford Musculoskeletal Biomedical Research Unit, Oxford,,United Kingdom Oxford | Oxford University Hospitals NHS Trust, Oxford, United Kingdom | Medical Diagnostics (United Kingdom),Yarnton, United Kingdom |

Table 6: **Comparison between our model and TokenSim**. Our model makes a correct predictions in these examples and TokenSim makes incorrect predictions. We observe that our model seems to identify salient tokens to correctly make these predictions.

For MAP and Acc. at K for all k, we notice that our model (or a variant of it) has the highest mean across the different features used except for Acc. at 50 when using the institution name and location only. When using all the features, the S-Trans which learns a linear combination of the values $\langle f(t_{s_2}), (t_{s_2} \cap t_{s_1})\rangle$, $\langle f(t_{s_1}), f(t_{s_2} \cap t_{s_1})\rangle$, $\langle f(t_{s_2}), f(t_{s_2} \setminus t_{s_1})\rangle$ with the the weights initialized to its default initialization is the top performing. Note that the S-Trans which uses the default initialization (our proposed model) outperforms the S-Trans which just learns a linear combination of the values. We hypothesize that randomly initializing the weights causes the model to get stuck in a bad local optima, while initializing the weights to its default initialization helps it find a better local optima. Lastly, we observe that in general using all the features (the institution name, location, and type) increases the performance of all the models.

To further evaluate our model, we perform 5-fold cross validation rather than using the proposed split (Table 4). Though the average MAP for O-LSTM is slightly higher than the average MAP for S-Trans, the standard deviation is 8 times as high. We note that S-Trans performs best in terms of Acc.@1.

### 4.1 Analysis

**Performance By Token Overlap**   We hypothesize that predicting `sub-institution-of` between institution strings that share a large number of tokens is easier than when fewer tokens are shared between the strings. To analyze this hypothesis, we partition the test examples into buckets by percent token overlap between the query and the true positive (i.e., the first bucket contains pairs

| Descendant Entity | Ancestor Entity |
|---|---|
| London Chest Hospital,London,,United Kingdom | Barts Health NHS Trust,London,United Kingdom |
| Pharmacy and Poisons Board | Ministry of Health |
| BASF (Canada),Mississauga,Ontario,Canada | BASF (Germany),Ludwigshafen am Rhein,Germany |
| BASF (China),Shanghai,,China | BASF (Germany),Ludwigshafen am Rhein,Germany |

Table 7: **Challenging Examples**. These examples highlight the difficulty of predicting sub-institutions. There are examples with very little textual overlap and examples with drastically different geographic locations.

| Descendant Entity | Ancestor Entity | SetTransformer Rank |
|---|---|---|
| Belmont Hospital, Belmont, New South Wales, Australia | New South Wales Department of Health, North Sydney, New South Wales, Australia | 7 |
| John Hunter Children's Hospital, Newcastle, New South Wales, Australia | Government of New South Wales, Sydney, New South Wales, Australia | 2 |

Table 8: **Discovering Missing Edges in GRID**. These are examples of pairs of institutions do appear exhibiting `sub-institution-of` which are not in GRID (i.e., false negatives).

with 0-5% overlap, the second 5%-15% overlap, etc.). Figure 3 contains the result. We observe that MAP scores for all models increase with token overlap. We observe that S-Trans has a larger margin of improvement over O-LSTM and TokSim when the amount of token overlap is smaller. We hypothesize that the set-difference-base term in S-Trans objective are useful in determining which non-overlapping tokens are meaningful when making `sub-institution-of` predictions.

**Examples** We provide qualitative examples of `sub-institution-of` predictions by our proposed model as well as by other baseline methods. These examples are given in Tables 5 and 6. In Table 5, we compare the predictions of S-Trans and of O-LSTM for two queries. In prediction of the super-institution of *Northampton General Hospital* we observe that O-LSTM selects a healthcare service at the lexically similar *Northamptonshire*. We hypothesisize that S-Trans is less likely to make such a mistake because it would consider *Northamptonshire* in the group of non-overlapping tokens in the two strings and could penalize not matching such a specific word. Table 6 provides a similar comparison between S-Trans and TokSim's predictions.

Table 7 show examples that underscore the difficulty of institution hierarchy prediction. These examples are cases for which external data or contextual features are required to determine the correct relationships. For example, without knowledge that *BASF* is headquartered in *Germany*, it is difficult to predict that *BASF(Canada), Mississauga, Ontario, Canada* is sub-institution of *BASF(Germany), Ludwigshafen am Rhein, Germany*. Newswire, web data, or affiliation strings of authors could contain the required evidence.

**Missing Sub-Institutions in GRID** To determine the usefulness of our model and understand the quality of our labeled data, we look for false negatives, i.e, pairs of institutions that exhibit the `sub-institution-of` relationship, but are not labeled as such in GRID. We consider 50 institutions and consider the model's highest ranked predictions for these super-institutions. Among these, we found two examples of false negatives that are shown in Table 8.

## 5. Related Work

The proposed task in this paper sits at the intersection of learned string similarity methods, relation prediction and extraction, and more generally set-based neural network models.

A fundamental component of the proposed approach is to use a parameterized method for comparing the spelling of two institution names. Parametric string similarity methods have a long history. Classic methods (Levenshtein, Longest Common Subsequence, Needleman-Wunsch [Needleman and Wunsch, 1970], and Smith-Waterman [Smith and Waterman, 1981]) measure similarity via insert, edit, and delete operations used to transform one string into another. Other work uses generative models [Dreyer et al., 2008, Andrews et al., 2012, 2014, Faruqui et al., 2016, Rastogi et al., 2016] or conditional random fields [McCallum et al., 2005] to model the string mutation sequences in an effort to learn a parameterized method for similarity within a domain. Recently, Gan et al. [2017] proposed a neural network model based on CNNs applied to character n-grams to model string similarity.

Relation extraction systems such as OpenIE, Universal Schema and others [Angeli et al., 2015, Verga et al., 2016, Das et al., 2017] are often interested in predicting relationships such as *subsidiary* or other similar relationships to that of this paper. However, these systems use natural language context such as sentences or paragraphs to make predictions rather than the strings themselves.

Set-based embedding models has also been well studied [Ravanbakhsh et al., 2016, Zaheer et al., 2017, Ilse et al., 2018, Hartford et al., 2018, Lee et al., 2019, Mena et al., 2018, Cotter et al., 2019, Bloem-Reddy and Teh, 2019]. Much of this work focuses on learning representations for sets that are order invariant. Our proposed approach models tokens in the intersection / set difference of the institution names, making set-based models a natural choice as our encoder's architecture.

## 6. Conclusion

In this paper, we consider the task of predicting institution hierarchies using a model-based on set transformers using institution names and other metadata. We hope that our work, which includes a dataset built from GRID, spurs interest in this challenging and important problem and that future work explores both richer spelling-based, context-free models as well as models that make use of natural language text and other contextual resources. In addition, rather than constructing set intersections by finding overlapping tokens, we will consider model-based methods for constructing set intersections (i.e., functions of geometric representations, such as Box Embeddings [Vilnis et al., 2018] or Order Embeddings [Vendrov et al., 2015]).

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
