# OpenReview forum: "Predicting Institution Hierarchies with Set-based Models"
_AKBC.ws/2020/Conference — AKBC 2020_

### Official Review · AnonReviewer1 · 2020-03-16
**New task, dataset and model**

**Rating:** 6
**Confidence:** 4

**Review:**

The paper addresses the prediction of the hierarchical structure of organizations / institutes. The authors develop a new dataset, automatically derived from GRID (global research identifier database), and compare a set-based model against a few baseline approaches. While the task is well-defined and the dataset could potentially be interesting for the community, I have a few doubts regarding the experimental setup (to be more specific: on the choice of baseline models, on the evaluation on the test set and on the final results).

Pro:
The task sounds interesting and challenging. It could encourage researchers to build and enhance models that combine knowledge from different sources.

Con:
The task is presented as a knowledge base completion task. Under this premise, I would have expected (1) a comparison with well-known baseline models from the knowledge base completion literature, such as TransE, and (2) a manual evaluation of the results on the test set. However, the authors mention embedding-based approaches only briefly as future work and evaluate the results only based on the manually extracted test set (which is very likely incomplete).
Furthermore, given the very low numbers (model results) and the examples in Table 3, it is questionable whether the proposed token-based models are a promising choice for the task.

Further comments / questions for authors:

- Equation on page 4: I read your motivation for the subtraction. However, I wonder whether your assumptions always hold (for example, would <f(t_s_1), intersection> - <f(t_s_2), intersection> mean that s_1 needs to contain more tokens than s_2?). Have you considered just feeding the different dot products into a feed-forward layer to allow the model to determine the best function automatically?

- Why do you use character-representations for countries but token-representations for cities and states?

- Page 6: in your itemized list of models, S-Trans is your proposed model and not a baseline model, right? This does not become very clear at that point.

- When you use another scoring function for some of your baseline models, the comparison among models is not valid anymore. Can you do an ablation study/analysis on your model, using different scoring functions as well?

- Table 3: These examples show that string-based models are not sufficient for the task. It would be very valuable to add an embedding-based baseline model to the paper.

- After looking at the results in Tables 4 and 5, I'm wondering how useable the approach would actually be for knowledge base completion and whether it would make more sense to use TokSim for that downstream application instead of S-Trans (because of the higher hit@1 score)

- Page 8: typo: in the last sentence before Section 4.1, there is a word missing (maybe "outperforms"?)
- Page 10: typo: BASG => BASF

- The related work on set-based models should be extended.

---

> ### Author Response · Authors · 2020-04-18
> **Response to Reviewer 1**
>
> Thank you for your time and consideration of the paper as well as your feedback. We’ve responded to your questions below.
>
> Expected comparison to TransE: TransE and related models only make predictions for entities that were seen at training time. They do not offer a mechanism to generalize to unseen entities. In our case, there is  _no intersection_ between the set of training organizations and test organizations. Our work only considers 1 relation type ---is-ancestor--- which inherently limits the effectiveness of techniques like TransE, which rely on multiple relation types.
>
> Manual Evaluation of Test Set: We believe that a manual evaluation would only marginally improve the quality of the paper since we believe that GRiD is relatively accurate and we derive our labels from there. However, we suspect that there may be False Negatives in GRiD (i.e., incomplete hierarchies), and so we will manually inspect a sample of test cases and add the result to the paper for a camera ready.
>
> Clarification of future work using embedded models: The approach presented in this paper represents sets as embeddings using Set-Transformer based models, and is thus an embedding based model (rather than a token-level method). We will clarify the comment about future work: rather than constructing set intersections by finding overlapping tokens, we will consider model-based methods for constructing set intersections (i.e., functions of geometric representations of entities, such as Box Embeddings (Vilnis et al, 2018) or Order Embeddings (Vendrov et al 2015)).
>
> Manually extracted test set is likely incomplete: first, we emphasize that our test set is extracted automatically. Second, we are confused by the term “incomplete.” To clarify: no partial hierarchies exist in the test set, i.e., if an entity appears in the test set, so does the entire hierarchy of which it is a part.
>
> Equation 4: First, let us clarify that s1 does not need to have more tokens than s2 as we allow empty set intersections in the model. We have run an additional experiment where we feed the three dot product scores as input to an MLP and a learned linear model. We found that neither of these models is significantly better than our proposed model. We have updated the paper with the new experimental results.
>
> Character vs Token Level Location Features: We ran additional experiments where we compared using character level models and token level models for state, country, and types. We have updated the paper with these results. We observed that the character level models generally outperformed the token level model. We noticed that in several examples where the character level models made correct predictions and the token level models made incorrect predictions that the locations of the institutions were out-of-vocabulary (unseen at training time). We hypothesize that character level models have better performance because they more gracefully handle unseen locations.
>
> Clarification of name “S-Trans”: We have updated the paper text and tables to make it more clear that S-Trans is the proposed model in this paper.
>
> Ablation study: This paper compares two general classes of models. The first (and proposed approach) uses a set-based scoring function that considers token intersection and difference between the two strings. The second embeds each string and models the hierarchy by way of similarity of embeddings. The proposed approach uses the set-based scoring function with a set-transformer (Lee et al, 2018) based encoder. For the first class of models, we perform an ablation study that replaces the set-transformer with a simple average of embeddings. We have also added experimental results with various scoring functions (e.g., linear model, mlp, etc.). We compare two architectures for the second class of models (LSTMs and transformers with position embeddings).
>
> Embedding based baseline: Our method is embedding based, we use a model that parametrizes each word with a vector and models sets of vectors as embeddings as well. We include baselines that are embedding based as well including an LSTM and transformer baseline.
>
> Insignificance of the results: we believe that the models presented in our work can be helpful in candidate generation for organization hierarchy-related tasks (e.g., manual curation and labeling, etc.). Note that hits at 1 is, in general, insufficient for evaluating our models because our models predict the is-ancestor relationship, and the vast majority of organizations we consider have more than one ancestor.  At a higher level, the precision achieved by our models is a testament to the difficulty of this task.
>
> Typos: we have corrected these, thanks!
>
> Related work: Thank you for your comment, we have added additional related work to the paper.

---

### Official Review · AnonReviewer2 · 2020-03-24
**Nice application but low originality and weak empirical evaluation**

**Rating:** 5
**Confidence:** 4

**Review:**

The paper shows how to infer the organisational structure of an institution. That is, it presents a model for predicting the is-ancestor relationships of institutions based on their string names. To this end, it makes use of Set-Transformers to model the token overlap between the institution names. This use is nice but also not highly original. The experimental evaluation is on a single dataset only. While the authors do present some examples, and overall hierarchy or something that provides some more insights into the learned model should be provided in order to show potential issues with transitivity and connected components. The evaluation only considers known pairs. But an organisational structure should also be consistent.  That is, the interesting motivation provided in the intro is not met in the experimental evaluation. Furthermore, the experimental protocol is  unclear. It seems to be a single training/test split, although one should consider several random splits or even some form of cross-validation, i.e., over the connected components given.

---

> ### Author Response · Authors · 2020-04-18
> **Response to Reviewer 2**
>
> Thank you for your time and consideration of the paper as well as your feedback. We have responded to your questions below.
>
> Transitivity: thank you for the comment about using our model to predict institution hierarchies. Note that our models rank candidate entities rather than making yes/no classifications, and so hierarchy construction is not the focus of this work. Previous work on predicting hierarchical relationships between entities also uses ranking-based evaluation of the is-ancestor relationship, for instance (Nickel & Kiela, 2017, https://arxiv.org/pdf/1705.08039.pdf). Our expectation is that practitioners can use our models to filter large collections of institutions to facilitate hierarchy construction and curation by other methods including manual effort. While predicting is-parent may make hierarchy construction easier, it may be more difficult to model since many institutions and their non-parent ancestors have highly similar names. We believe that trying to correctly separate parent institutions from ancestor institutions along with performing classification instead of ranking could be an interesting path for future experiments.
>
>
> Cross Validation: cross validation would strengthen the results. We will do this for the camera ready. For now, we are reporting means and standard deviations achieved over 5 different random seeds. We have clarified our experimental setting in the paper.

---

### Official Review · AnonReviewer3 · 2020-03-30
**Good work with some ambiguity and weakness in contributions and presentation**

**Rating:** 7
**Confidence:** 4

**Review:**

Paper is on modeling the prediction of ancestor relation between names of science institutions.  This is on the GRID dataset which already has some hierarchical information.  The proposed approach is set-based models (with neural encodings) where the overlap between two names is measured by set overlap at the unigram level.  In extended experiments additional metadata like address and type of institution are also incorporated into the model (which contribute a lot to the improvements).  A set of simple to intermediate baseline along with different thresholds of token overlap has been tested and the proposed model shows strong improvement in the MAP metric.

Paper has a decent writing and structure.  Problem and the approach has been explained and motivated well with descriptive examples.  The proposed approach has fair amount novelty in using the set-based encoders with a transformer architecture which shows promising improvements against convincing baselines.  However, some major aspects of the contribution is not explained well.  For example, from abstract and intro, it seems that the dataset is a major contribution of this work, while it is not clear what's the actual dataset? Moreover, it is not clear if the test set is gold-standard here (specially the negative examples that seem to be constructed heuristically).  If one premise of the work is to correct noise in GRID, then the same (potentially noisy) data can not be used as test set.

Some questions and suggestions:

1. What's the nature of the data that you've constructed?  Is it pair of names with binary classes?  Has a human reviewed this data (specially the heuristically constructed negative instances)?  How many names?  Are you releasing all of it or subset of it?

2. One premise of the work is the usage of set-based models.  There, don't you think unigram-level overlap measurement is just too simple?  distinguishing many ancestor orgs in "washington university" vs. "university of washington" can be quite ambiguous on a unigram-based set models (without using meta data like address).  Did you think of expanding to at least bigrams in some non-expensive ways?

3. Some Tables are not descriptive neither by the caption nor the description in the body.  For example in Tables 1-3 (which have almost same long caption) what are those names under the baseline columns?  Probably those are (incorrect) predictions of the baseline model?  And probably the proposed model (e.g. set transformers) does not predict those.  This needs to be clarified and not guessed.  Or in Table-4, where different kinds of GRID features are getting incorporated, the description of the experiment is 2 pages earlier and caption is quite non-descriptive.  And caption of Table-5 is plain non-informative.

4. there is little information on the the set encoder was actually trained.  is it exact replica of Lee et al. 2018?

5. in table 4, why there's such major drop in the O-Trans column (is 0.01 a typo)?  And why addition of the  "type" meta-data doesn't help?  is it noisy?

6.  Given the fairly low level of precision (even for the proposed approach), how can one actually use this?

7. Minor thing: In constructing your data, how do you deal with institutions that have multiple parents (e.g. Lawrence Berkeley Lab which is part of UC-Berkeley and US department of Energy.  Are there multiple instances the child inst.?

---

> ### Author Response · Authors · 2020-04-18
> **Response to Reviewer 3**
>
> Thank you for your time and consideration of the paper as well as your feedback. We have responded to your questions below.
>
> Nature of the data: The data is generated directly from the GRiD xml database dump. We will release the data we used in the paper. We have inspected, but not exhaustively inspected the constructed negatives. The training data is a list of triples of (node, ancestor_node, non_ancestor_node) examples. The dev and test data are pairs that are (node1, node2, ancestor_label). Negative examples for an organization are constructed by selecting all non-ancestors within the organization's hierarchy as well as random non-ancestor organizations from outside the organization’s hierarchy that share a city, state, or country. Negative examples are restricted by their train/dev/test partition so that the model does not see any of the negative examples shown at training time at dev/test time.
>
> Set based models: we think that set based models are vital to this task because we do not believe that order matters (very much) when predicting institution ancestors. We agree with your observation that unigram-level token overlap may result in ambiguity, but we also believe that measuring overlap in bigram-level token overlap would result in similar ambiguity (examples). However, we believe that this is an interesting avenue for further experimentation--thank you for the suggestion!
>
> Tables: we have now made this clearer in the paper.
>
> Replica of Lee: Yes it is a replica, though since our sets are small in size we do not use the inducing point method given by Lee et al to scale to large sets.
>
> Drop in O-Trans column:  In investigating our performance, we found a bug where positive examples were also labeled as negative examples, and retrained our models and updated the paper. Now, the O-Trans model’s performance increases as the model has more features.
>
> Value in practice: while the performance of our method is far from perfect, we believe the hits@k results for higher values of k are reasonable.  We envision using our models to filter the candidate ancestors for a given organization,  and using additional tools (models, manual effort) to curate the filtered set and/or use it to construct a hierarchy. Given the difficulty of this task, we think that this is the most effective means of creating accurate institution hierarchies, providing a step towards automated institution hierarchies.
>
> Multiple parents: we do not treat institutions that have multiple parents differently than institutions that have have a single parent. All parents (and those parents’ ancestors) of a given institution are considered ancestors of a given institution.

---

### Decision · Program_Chairs · 2020-05-01

**Decision:**

Accept

**Comment:**

This paper presents a new task and model for predicting the hierarchical structure of organizations / institutes. The model predicts is-ancestor relationships between the institutions by modeling the set operations between the strings. The authors develop a new dataset, automatically derived from GRID, and compare their set-based model against a few baseline approaches.

The reviewers’ comments were generally positive and praised the usefulness of the task, which makes us recommend acceptance. However, there were some concerns about the experimental setup (choice of baselines and evaluation).   We strongly recommend improving these aspects on the final version, as per the reviewers’ suggestions.